# Evaluation of Fetal Exposures to Metals and Metalloids through Meconium Analyses: A Review

**DOI:** 10.3390/ijerph18041975

**Published:** 2021-02-18

**Authors:** Stephani Michelsen-Correa, Clyde F. Martin, Andrea B. Kirk

**Affiliations:** 1AAAS Science & Technology Policy Fellow Hosted by EPA Office of Chemical Safety and Pollution Prevention, Biopesticides and Pollution Prevention Division, Washington, DC 20004, USA; smcorrea@uw.edu; 2Department of Mathematics and Statistics, Texas Tech University, Lubbock, TX 79409, USA; 3Department of Occupational and Environmental Health, Milken Institute School of Public Health, George Washington University, Washington, DC 20052, USA; abkirk@gwu.edu

**Keywords:** heavy metals, meconium, fetus, fetal exposure, biological monitoring

## Abstract

This paper surveys the existing scientific literature on metals concentrations in meconium. We examine some 32 papers that analyzed meconium for aluminum, arsenic, barium, calcium, chromium, copper, iron, lithium, magnesium, manganese, zinc, lead, mercury, manganese, molybdenum, nickel, phosphorus, lead, antimony, selenium, tin, vanadium, and zinc. Because of the lack of detail in the statistics it is not possible to do a rigorous meta-analysis. What stands out is that almost every study had subjects with seemingly large amounts of at least one of the metals. The significance of metals in meconium is not clear beyond an indication of exposure although some studies have correlated metals in meconium to a number of adverse outcomes. A number of outstanding questions have been identified that, if resolved, would greatly increase the utility of meconium analysis for assessment of long-term gestational metals exposures. Among these are questions of the developmental and long-term significance of metals detected in meconium, the kinetics and interactions among metals in maternal and fetal compartments and questions on best methods for meconium analyses.

## 1. Introduction

Chemical manufacturing has seen a 10-fold increase in production over the last 40 years [1]. Exposure to environmental contaminants, including heavy metals, is a growing public health concern as global chemical production continues to increase. Pregnant mothers and children exposed in utero and early childhood are particularly susceptible to the harmful effects of chemical pollutants [2,3,4]. Excessive exposure to metals during fetal development leads to a wide range of adverse outcomes. The effects of metals exposure may be apparent at birth [5] or emerge later in childhood or even adult life [6]. For example, fetal exposure to some metals is linked to congenital heart disease [5,7]. Other problems associated with fetal similar exposures that may become apparent in childhood include increased risk of attention deficit hyperactivity disorder (ADHD) [8] and other behavioral or intellectual deficits [9], type 1 diabetes [10,11] and immuno-dysregulation [12]. Methods to determine prenatal heavy metal exposure are needed for assessing longer-term gestational exposure to heavy metals and the risk of adverse human health effects later in life.

Fetal metal exposures are typically gauged by analyzing maternal and cord blood. Results of maternal and cord blood metals analyses indicate recent maternal exposures (unless mobilized from maternal stores i.e., lead from bone) and indicate fetal/infant exposure around the time of birth. To gauge longer-range exposures, researchers have analyzed metals in hair, nails, and meconium. Meconium analysis has been used to evaluate fetal exposures to drugs [13], nicotine [14], pesticides [15], and as addressed in this paper, heavy metals. While there are many studies of metals in meconium, there has not yet been a systematic collection and review of the available data. As such, it is difficult to differentiate “background” metals exposure from specific points of exposures (i.e., mines, industrial activity, etc.), or to determine concentrations indicating a detrimental exposure.

Meconium is a thick, dark, viscous material comprised of accumulated waste material ingested with amniotic fluid (AF) during the second and third trimester of pregnancy. Meconium is normally expelled by the newborn during the first day or two following parturition. Expulsion prior to birth is considered a sign of fetal distress. Because meconium represents accumulation of material over approximately six months, its analysis can be a useful tool in assessing longer-term gestational exposure to heavy metals. While analyses of maternal blood, cord blood and AF can be used to assess exposures, they are, to a greater degree, limited by the point of time of collection. Analysis of meconium for metals has been used to assess longer-term (second and third trimester) fetal exposures among women living in highly polluted environments [16].

Meconium analyses for metals have been used to explore relationships among toxic metals and birth defects including heart [7] and neural tube defects, prematurity fetal growth retardation, prematurity, pre-eclampsia [17], gestational diabetes, and neonatal loss [18]. In the case of essential minerals, meconium analyses have been used to assess nutritional adequacy. Meconium analysis is also used to evaluate fetal exposures to drugs [19], nicotine [20], and pesticides [15]. Fetal metal exposures have been gauged by examining maternal and cord blood. Results of maternal and cord blood metals analyses likely indicate recent maternal exposures (unless mobilized from maternal stores such as lead from bone) and provide an indication of fetal/infant exposure around the time of birth. To gauge longer-range exposures researchers have also analyzed metals in hair, nails, and meconium. While there are many studies of metals in meconium there has not yet been a systematic collection and review of the available data. As such, it is difficult to differentiate “background” metals exposure from specific source exposures (i.e., mines, industrial activity, etc.), or to determine concentrations indicating a detrimental exposure.

Essential metals such as zinc (Zn), selenium (Se), iron (Fe), copper (Cu), magnesium (Mg) and manganese (Mn) are needed for the synthesis of cofactors serving a wide range of processes during fetal development and in maintaining homeostasis throughout the lifespan. Other metals having no known biological use are detrimental to human health at all levels include arsenic (As), mercury (Hg) and lead (Pb). We note that some metals, such as Cu, Zn and Fe have two roles in that they are essential nutrients but have detrimental effects in excess. Some metals such as Hg are highly toxic when organified (i.e., methylmercury). Tin is not nutritionally necessary and as an element has low toxicity. However, fetal exposures to organotins (i.e., tributyltin) can have multi-generational health effects and lead to metabolic disorders [21]. The studies surveyed in this paper used analytical methods that allowed reporting of metals but did not distinguish between organometals and other forms. While highly important, organometals are not discussed in this paper in any detail.

## 2. Methods

A literature search was conducting using AbstractSifter [22] and PubMed query run: Heavy Metals OR Arsenic OR Hg OR mercury OR Pb OR cobalt OR copper OR zinc OR cadmium AND meconium. The bibliographies of the collected papers were then reviewed and any that reported on metals in meconium, with the exception of iron, were added. The methods and results sections of each paper were examined for errors or omissions. Following this methodology, 34 papers were selected for data extraction. Information was collected on year of publication, analytical method, cohort size, cohort comparative attributes (male vs. female, birth defects vs. no defects, premature infants vs. term, diabetic vs. non-diabetic mothers, and dizygotic vs. monozygotic twins) location of cohort, any notable industrial, mining or pollution concerns and any data on meconium metals concentrations.

## 3. Results

The query identified 148 papers with publication dates ranging from 1964 to 2019. Titles and abstracts of the papers were reviewed for relevance (were about metals in meconium at least in part). Studies of meconium aspiration or other clinical syndromes or reports of analyses that did not contain data on metals were excluded from further review. The bibliographies of the remaining papers from the query were reviewed and any referenced papers containing data on metals in meconium were obtained and relevant data extracted. The 34 selected papers contained data from approximately 7000 infants and 73 distinct cohorts. No papers containing data on metals in meconium were excluded for methodological reasons. The studies varied by sample preparation method, analytical technique and data reporting styles.

### 3.1. Analytical Methods and Reporting

Analytical methods used to determine metals concentrations included Atomic Absorption Spectrometry, Inductively-Coupled Plasma-Mass Spectrometry, Direct Mercury Analyzer, various colorimetric methods, Graphite Furnace Atomic Absorption Spectrometry, Cold Vapor Atomic Fluorescence Spectroscopy, and Inductively Coupled Plasma Atomic Emission Spectroscopy. Sample preparation methods included acid digestions, dry-ashing [23], or were not discussed in detail [24,25]. Most, but not all, of the protocols involved drying samples prior to sample preparation and analysis. For a few studies it was unclear if samples were dried prior to analysis [26], and one study reported drying samples to “near dry” [27]. Metals were reported, with two exceptions, in parts per million (ppm), billion (ppb) or trillion (ppt). For the purpose of comparing study results, all of the units were converted to ppm as this was the most common unit reported. The excepted studies [23,25] reported total metal in each sample per kg of infant body weight (ug/kg) or ug/g/kg of infant body weight without reporting the weights of the infants. Our ability to compare or otherwise analyze the results of the pooled data is constrained by these differences. It is important to note that one study ‘s non-detect might have been another study ‘s ppb or ppt. In addition, some studies reported the percent of samples containing detectable metals followed by report of their either their mean, median, range, or interquartile range. For example, [27] report a mean meconium cadmium (Cd) concentration of 13 ppm. However, their mean omits the 91.5% of samples whose meconium contained Cd below the limit of detection. Lead levels were similarly reported. [20] reported a mean Pb concentration of 120 ppb with quantifiable lead in only ~26% of samples. The 120 ppb mean was approximately 10× greater than levels reported by others and omission of non-detects may explain the difference. Other studies reported results below the limit of quantitation or non-detects as “zero” and reported means and standard deviations that included these values without reporting data on individual samples.

The studies surveyed used analytical methods that allowed reporting of almost exclusively elemental metals. These metals may have been present in meconium as organometals but analytical methods did not allow for their analysis.

### 3.2. Description of Cohorts

There was a great deal of variation in the description of the mothers and the infants. Some studies reported minimally on their subjects [28] or designed their studies to include only “healthy” or “term” infants. Other studies were designed to compare two different populations (premature, heart defects or gestationally diabetic mothers) with controls. Only one study [29] recorded maternal use of prenatal vitamins.

The details are summarized in the Table 1 below.

### 3.3. Premature vs. Term

Among the four studies comparing metals in meconium of premature vs. full-term infants, [28] reported higher concentrations of Al, Cr, Cu, Fe, Mn, Mo, and Pb in the meconium of premature vs. full-term infants with statistically significant differences reported for Fe and Cu and concentrations of Zn and Mg (both essential metals) higher in term infants, but not statistically significantly. [23], in an analysis of Cu, Cr, Fe, Mn, and Zn levels in term and premature infants, observed lower concentrations of these metals in premature infants. Data for the metals that the four studies analyzed in common are shown in Table 2. Data are reported as comparatively higher or lower by birth status rather than with a data analysis because of differences in data reporting, analytical methods, and sample collection techniques.

### 3.4. Statistical Approaches

Basic statistics of data analysis are most often concerned with two things; the center of the data and the spread of the data. The center of the data is measured by calculating one of three numbers, the mean, the median or mode. The mean is simply the numerical average; the median is the number chosen so that one half the data is numerically less and the mode is the number that represents the most common value. The field of order statistics uses the median as its most common measure of central tendency. In what will here be referred to as standard statistics the mean is the most commonly used measure of central tendency. For large data sets the two measures tend to approximate each other. In general, it is not possible to compare the two measures except to say they are both important measures of central tendency, especially for relatively small data sets such as those being discussed in this survey. Types of statistics used in various papers are shown in Table 3. It would be advisable for studies that have a relatively small number of subjects to use order statistics in their reports.

Metals analyzed most and least frequently by researchers are shown in Table 4 and Table 5.

Metals detected in meconium:

Aluminum (Al): Studies of the effects of gestational Al exposure were sparse compared to some other metals. However, detailed but perhaps dated human health risk assessments for Al [52] and species exist [53]. Early exposure was associated with neurological impairments [54] possible endocrine disruption [55] as well as increased placental oxidative stress and inflammatory response [56].

Four studies (Table 6) reported on Al in meconium with the mean of means equal to 25 ppm [15,28,30,37]. All studies except [30] reported means and standard deviations and all but [28] used an ICP-MS based analytical method. Reported means ranged from 6.577 to 43 ppm. Aluminum levels were higher among infants born to women living in industrial areas [30] and among those born to diabetic mothers [37]). Aluminum exposure was associated with diabetes [57].

Arsenic (As): Arsenic is a toxic metal that has no known biological role. It is associated with various cancers including lung and skin. It occurs as a byproduct of lead and Cu smelting and was formerly used as a wood preservative. Fetal As exposure can cause changes in the placental transcriptome and is associated with reduced birth weight, especially in female fetuses [58] and alterations in glucocorticoid system [59]. Arsenic is also associated with increased risk of gestational diabetes [10,60].

Seven authors tested for As in meconium with 13 cohorts. The average mean reported was 0.0479 ppm with a standard deviation of the distribution of the means of 0.0401 ppm. The means found were remarkably close. Several studies reported that the results were below their detection levels. Still, as an indication of exposure it is concerning that As was found. The presence of As in meconium is not unexpected, as it is a common water contaminant in some regions and when present in soil or irrigation water, it can accumulate in produce. Arsenic underwent primarily urinary excretion [61] and only a relatively small proportion of a dose was found in adult feces. For the fetus, handling of As would be quite different. AF movement occurs primarily between the fetus and the amniotic fluid as the fetus swallows and urinates. For a fetus, direct urinary excretion to the external environment is not an option.

Barium (Ba): Only one group included Ba in their analyses [37] and reported levels twice as high in meconium from infants of diabetic vs. non-diabetic mothers (0.3 ± 0.16 ppm vs. 0.15 ± 0.19 ppm). Fetal exposure may increase the risk of congenital heart defects [62], and orofacial clefts [63]. Adult exposures are associated with hearing loss [64] and diabetes [65].

Cadmium (Cd): Cadmium is found in the earth ‘s crust, is associated with Zn, lead, and Cu ores. Cadmium is released to the environment through manufacturing and the burning of coal, fossil fuels and waste. People living in proximity to such pollution sources have higher Cd exposures than people living elsewhere [66]. However, Cd becomes widely disbursed through atmospheric processes, where it is bioconcentrated by plants, enters the food chain, is inhaled (cigarette smoke), or incidentally ingested along with contaminated soil. Eight studies included tests for Cd. Calculating the average of the means yields 1.52 ppm with standard deviation of 2.93.

Chromium (Cr): Four studies analyzed for Cr [15,28,36,37]. Study means ranged from 0.03–3.2 ppm. The highest concentrations of Cr were from 12 premature infants in the [28] study. Exclusion of their data left the mean of means at 0.29 ± 0.58 ppm. Chromium is an essential nutrient that can be highly toxic when intake is in the form of CrVI. As with other metals data were reported as total Cr. It is impossible to know at this point if the Cr in meconium reflects intake of an essential nutrient, or if it reflects a potentially dangerous fetal exposure.

Cobalt (Co): Cobalt is an essential nutrient required for synthesis of vitamin B12. Deficiencies can cause anemia and neurological deficits. Three studies analyzed meconium samples for Co [15,37,51]. Levels reported by [51] were substantially higher than those of [15] and [37] two cohorts (see Table 7). B12 for the most part undergoes fecal excretion so the Co reported may be part of normal Vitamin B12 kinetics. Or it might indicate fetal exposure to environmental Co.

Copper: Copper is an essential nutrient required for the functioning of numerous enzymes including ferro-oxidases that are required for binding Fe to transferrin, an Fe -binding distributor protein. Copper deficiency can therefore lead to Fe deficiency. Because it causes oxidative damage in its elemental ionic form and is also an essential nutrient, Cu is tightly regulated. Copper is sequestered by chaperone proteins, stored for future use when not of immediate need, and transferred through carrier mediators [67,68,69]. High affinity Cu uptake protein 1(Ctr1) is expressed in placenta [70] as are a pair of Cu transport ATPases: ATP7A (also known as Menke ‘s protein) which transports Cu to fetal circulation, and delivers it for synthesis of cuproenzymes. ATP7B is expressed (or translocated to) the placental microvilli when Cu levels are in excess and transport Cu back to the maternal circulation. Unless there is a mutation (as seen in Menke ‘s disease, or the mother was exposed to exceedingly high levels of Cu, it seems unlikely that that a fetus would be exposed to dangerous amounts of Cu. Copper deficiency is thought to be quite rare, although a risk for those who are exposed to, or ingest, high levels of Zn [71].

A total of 15 studies tested for Cu. In the studies we examined there was great variation in the amount of Cu found in meconium. The average level was 45.8 ppm and the highest cohort average was 117 ppm. The smallest nonzero mean, 0.01475 ppm, was recorded by [33] in their South Carolina cohort. The standard deviations were difficult to summarize but it was clear that most of the distributions were very long tailed on the right. In the set of studies by [24]. the standard deviations were all greater than the means which can only happen if there are large outliers on the right. Even the distributions of the means are skewed to the right with a standard deviation of 53.5. [23] cohort of Canadian term infants from Hamilton, Ontario was more than three standard deviations from the mean at 245 ppm and about twice as high as concentrations found in their premature infants from Newfoundland. [23] used atomic absorption spectrometry.

Lead (Pb): Lead is a well-recognized neurotoxicant. Exposures can have many sources including old lead paint, lead water pipes, dust and other point and non-point sources. Lead will accumulate in bone and may be released to the maternal blood stream during pregnancy, exposing her fetus, even if the mother is not actively exposed in her current environment. Lead is associated with congenital heart disease [5] among other things. There is no known safe level of exposure. A total of 13 studies tested for lead in meconium. The mean of means was 7.8 ppm with ranges from below the LOD to 603 ppm in a newborn from a highly industrialized area in the Philippines, [20]. Again, every cohort had extreme outliers, which indicates that local background levels are frequently not the main source of lead exposure.

Lithium (Li): Lithium in groundwater is common in some areas. It is also used as a therapeutic for depression and bipolar disorder. Lithium readily crosses the placenta and is considered a teratogen as well as an inhibitor of fetal growth. 

Only [37] included Li in their analyses of meconium from infants from diabetic vs. non-diabetic. Mean concentrations were nearly identical (2.429 vs. 2.498 ppm) and standard deviations were the same (0.4). Other metals from [37] were not so strongly matched. Detection of Li in meconium at this point simply tells us that these infants were exposed. [37] did not report any birth defects among the cohorts and found no association between Li exposure as measured through meconium and gestational diabetes.

Mercury (Hg): Mercury is another well-recognized neurotoxicant and is readily transfered from mother to fetus [72]. A total of 10 studies reported on findings of Hg in meconium. One study detected Hg in all samples but at the lowest level it was too low to quantify. The medians were reported for six cohorts with an average median of 0.0207 and a standard deviation of the medians was 0.0138. The average reported mean (ten cohorts) was 0.98 with one mean of 9.45 ppm from a cohort born in a highly industrialized city where local fish consumption is common strongly skewing results [34]. The [34] results may have been even higher than those of other studies because they did not report drying samples prior to analysis. [34] also provided mercury data from maternal and cord blood which were not correlated with meconium (r: 0.53; and 0.55 respectively). A later study [30] of newborns from the same city showed much lower concentrations of Hg in meconium with a reported median below the limit of quantitation (LOQ).

Magnesium (Mg): Magnesium is an essential nutrient but can be toxic at high levels. In mice, prenatal hypomagnesemia causes placental abnormalities, reduced fetal growth, higher postnatal mortality and failure to thrive for survivors [73] as well as neurological deficits [73]. High human prenatal Mg exposures may result in impaired bone mineralization leaving the newborn vulnerable to fractures [74,75]. In our review, we found six studies that tested for Mg in meconium. The average concentration found was 2665 ppm with the standard deviation of 1111. All of the studies had means within two standard deviations of the average mean. One study [28] found much lower concentrations of Mg and was almost two standard deviations below the mean. The 12 preterm and 38 full term infants in that particular study population may have been Mg deficient. However, this was one of the studies where it was not clear if meconium samples were dried prior to analysis. If meconium is 70–92% water [76], the comparative concentration of Mg in the [28] cohort may have been unremarkable.

Manganese (Mn): Manganese is an essential nutrient for mammals. It is common in the Earth ‘s crust and most vegetables have enough Mn to meet healthy dietary requirements. Vegetarians typically consume more Mn than omnivores. Manganese is believed to readily cross the human placenta [77]. There is growing evidence that excessive early Mn exposure results in adverse neurological [78] and behavioral outcomes later in childhood, particularly for girls and in those with polymorphisms in Mn transporter genes [79] The World Health Organization (WHO) has set a Mn standard of 400 ppb, although that level is often exceeded [80].

Manganese in meconium was addressed in eight studies for a total of 14 cohorts. The average mean for the 14 cohorts was 27.7 ppm and the standard deviation of the distribution of means was 18.5. All the studies were within the three standard deviation margin. In fact, no study lay outside the two standard deviation margin. Thus, the results from the 14 cohorts were remarkably close.

Molybdenum (Mo): Only [28,33] analyzed for Mo reporting 0.187 ± 0.112 ppm and 64 ± 22 ppm respectively. Only the [33] cohort (n = 15) from Butte, MT and not the cohort from South Carolina had detectable Mo. The area around and within Butte, MT was mined for Cu and other metals including Mo for around 100 years. These activities released other heavy metals to the environment. It is possible that the higher Mo seen in the Butte cohort was due to environmental contamination, but meconium tests of other metals were not exceptionally high. Mo is an essential nutrient important for synthesis of a number of enzymes including aldehyde oxidase, nitrogenase, and sulfite oxidase. Toxicity concerns were low, although high Mo intake could reduce the bioavailability of Cu.

Nickel (Ni): The presence of Ni in meconium of newborns from Sherbrook, Quebec, Canada was tested by [15] and in newborns from Xiamen, China by [37]. Concentrations of Ni in the [37] cohorts (diabetic vs. non-diabetic mothers) were nearly identical. Nickel is considered a low toxicity metal and is not nutritionally required. However, higher Ni exposure among pregnant women is associated with increased risk of preterm birth and low birth weight [81] and with congenital heart disease [37,82] reported no statistical difference between Ni exposure and markers of gestational diabetes. [15] reported lower levels (0.107 ± 0.081 with an interquartile range of below the LOQ to 1.7.

Selenium (Se): Selenium is an essential nutrient required for the synthesis of deiodinases and glutathione peroxidases and is a component of several amino acids. Low Se intake early in pregnancy is associated with premature birth [83]. Only [84] analyzed for Se finding a mean of 0.42 ± 0.46 ppm for 22 infants. Se intake may attenuate the effects of methylmercury [85]. Higher, but environmentally relevant, intake of Se may increase risk of diabetes mellitus [86] and amyotrophic lateral sclerosis (ALS) [87,88]. The association between diabetes and Se may be due to higher levels of Se transport protein selenoprotein P and not necessarily to higher Se exposure [89].

Antimony (Sb): Only [37] analyzed meconium samples for Sb, finding nearly identical concentrations in samples from diabetic vs. non-diabetic mothers (0.13 ± 0.09 vs. 0.12 ± 0.09) in Xiamen, China. Xiamen is considered a low-pollution city. However, China is the world ‘s largest Sb producer and researchers there have measured atmospheric deposition as particularly high in the areas surrounding Sb mines [90]. Antimony is used to harden bullets and has been dispersed into the environment through shooting training or recreational activities. In the US an estimated 1900t of Sb is thought to be released annually in the US alone [91]. Humans and wildlife may be exposed through contact with contaminated soil or through consumption of plants, which are known to take up Sb [92]. Other sources of exposure include antimony-containing drugs used for treatment of parasitic infections, but this should be rare.

Antimony is also transferred through milk [93]. A placental transfer efficiency of 80.1% was estimated by [49] for Sb. Juvenile exposures to Sb are associated with kidney damage in rats [94].

Tin (Sn): Tin is not nutritionally necessary. As an element, is not considered particularly toxic. However, this is not true for organotins, such as tributyltin, which can cross the placenta and accumulate in fetal tissues, primarily liver and brain [95]. Organotin exposures can result in multi-generational effects [21] and alter male [36] reproductive tract development in rat dosing studies along with hepatic steatosis, endocrine disruption and other adverse effects [96]. Only one study, a comparison in metals in meconium from infants of diabetic vs. non-diabetic mothers, provided data on Sn [37]. Infants from diabetic women had more total Sn (0.069 ± 0.063 ppm) than infants from non-diabetic women (0.043 ± 0.033 ppm). The study was undertaken in Xiamen, China, a port city with high levels of tributyltin in sediment and the tributyltin metabolite, monobutyltin, in seawater [97]. Consumption of local seafood may have been a source of exposure, but since there is no other population to compare Sn levels in meconium it is impossible to know how the Xiamen infants would compare to others. No study examined organotins in meconium, reporting only total Sn. Tin exposure is common in the US [98] and presumably in other countries as well.

Vanadium (V): Vanadium has no known biological function in humans. Prenatal exposure is associated with reduced fetal growth [99] and low birth weight [48,92]. There is also evidence for renal and hepatic toxicity [100]. Vanadium is used in synthesis of several drugs, some of which are used to control diabetes and other diseases [101] though they are not recommended for pregnant women). Vanadium exposures of industrial workers and laboratory animals are associated with adverse neurological outcomes [102] and with reduced fetal growth in a large epidemiological study [95].

Two studies measured V in meconium [15,37]. Both dried samples as part of sample preparation and analyzed for V by ICP-MS. Still, concentrations were markedly different, although this may be attributable to differences in handling non-detects or results that were below the limit of quantitation. [37] reported a mean of 0.0025 ppm for 137 infants of diabetic mothers and a mean of 0.0005 ppm for 197 infants of non-diabetic mothers. [15] reported a mean of 0.036 ppm for 371 infants from a metals mining area of Quebec.

Zinc (Zn): Zinc is an essential nutrient to all mammals and is of low toxicity. Following long term exposure, even at lower doses (~0.5–2 mg Zn/kg/day) of Zn compounds, the harmful effects generally result from a decreased absorption of Cu from the diet, leading to early symptoms of Cu deficiency. Insufficient Cu intake or Zn-induced Cu deficiency may lead to anemia. Copper is needed for synthesis of ferroxidase which binds Fe to transferrin which is needed for Fe uptake. When Zn is high, it can reduce Cu availability through induction of metallothionein. Copper complexed with metallothionein is retained in the mucosal cell, relatively unavailable for transfer to plasma, and excreted in the feces when the mucosal cells are sloughed off. Thus, an excess of Zn can potentially contribute to Cu deficiency [71]. Zinc supplementation or exposure to environmental Zn, might contribute to some of the proportionately higher Cu/low Zn ratios seen in some of the studies. If that is the case, then higher Cu in meconium may indicate greater risk of Cu deficiency and anemia.

For Zn the average value of the medians was 165 ppm with a standard deviation of 114. This indicates a distribution that seems fairly normal. The maximum median was reported by [15] at 314 ppm which was not quite two standard deviations from the mean and the lowest was reported by [33] at 0.053 ppm for their South Carolina cohort.

## 4. Discussion

Ionic metals may be transported from the maternal to fetal compartments by simple diffusion at the placental interface. Evidence for this is implied for some metals where is a close equivalence between maternal and fetal blood Pb [103]. However, not all metals show this association. Through similarities of charge, size and structure, some metals like Cd [104] can pass through channels whose biological function is to actively transport essential metals such as Ca, Fe or Zn [105]. While Cd may easily pass through essential metals transporters, it is also a strong inducer of metallothionein, which binds it and inhibits its passage to the fetus [106]. Cadmium concentrations are most commonly lower in cord blood than maternal blood and highest in the placenta.

The placenta plays a multi-leveled protective role against xenobiotic transfer to the fetus. The placental interface consists of two layers, the syncytiotrophoblast and the cytotrophoblast. The syncytiotrophoblast has a microvillous surface to increase uptake of blood-borne nutrients, and sometimes toxic metals or other deleterious agents.

Cells along the placental interface express a variety of proteins that sequester, detoxify or transfer toxic substances to the fetus or back toward maternal circulation. The ability of the placenta to manage transfer of metals (essential and non-essential) to the fetus may vary by trimester where differences in gene expression profiles have been noted [107]. Another factor is presented by differences in placental transporters over time. For example, MRP1is a transporter identified as playing a role in passage of methylmercury to the fetus [108].). MRP2 (ABCC2) is an ATP-binding cassette (ABC)-type membrane protein. It ‘s expression in the placenta varies by gestational age [109]. This may explain some of the differences seen in metals concentrations in meconium of premature infants vs. full-term infants, although it is certainly possible that they represent distinct exposure scenarios. 

Elemental metals: Metals showing predominance of urinary excretion include cesium, meconium, Co and selenium. Metals that primarily undergo fecal excretion, at least in adult animals include manganese, Cu, thallium, lead, Zn, Cd, Fe and methylmercury. Some metals, including As, elemental Hg and tin rely on both [110]. Metals may also deposit in liver [111], kidney [112] or other fetal tissues, in addition to deposition in meconium.

Metals excreted in urine may be re-ingested, and re-urinated by the fetus as excretion to the external environment does not occur. Metals with high solubility, can become trapped between the fetal compartment and AF as little fluid returns to maternal circulation across the amniotic membranes [113]. Unfortunately, few studies have looked at the potential for metals to accumulate in amniotic fluid over time. One study [114] compared heavy metals in AF between two different groups of women: 16 to 20 weeks (n = 19) and those between 21 and 26 weeks (n = 20). Significant differences in AF metals concentrations by gestational age were found only for Cu and magnesium, both essential nutrients. This was not the case for As or ten other metals and metalloids so there may be mechanisms for return to maternal circulation or to sequestration meconium or in fetal tissues. No other research on temporality of amniotic fluid metals concentrations were found. Temporal trends in meconium metal concentrations (proximal vs. distal or first vs. second expulsion sampling) have, as far as we know, not been investigated.

Organic Metals: While elemental metals may enter the fetal-placental-amnion compartment through passive means or through mimicry, organified metals may actively accumulate in the fetal compartment through transporters such as the Placental ATP-binding cassette (ABC) transporters [5,29] and the Large amino acid transporters 1 (LAT1-4F2hc) (*SLC7A5-SLC3A2*) and 2 (LAT2-4F2hc) (*SLC7A8-SLC3A2*). ABC transporters are present in placenta and both the apical and basolateral amniotic membrane [115] where they may play a role in active transport of methylmercury [116]. Methylmercury appears to be more easily transferred to the fetus than other, inorganic metals, most likely because of the relative ease with which methylated forms may undergo active transport [117]. There is evidence that Cd exposure results in down-regulation of some placental ABC transporters, possibly limiting Cd transfer to the fetus [118]. These transporters protect the developing fetus from xenobiotics, including drugs and pharmaceuticals and transport the essential mineral necessary for function and development. However, some toxic heavy metals including methylmercury [119,120] can use these routes to cross into the AF and then into the fetal compartment.

A final factor influencing metals deposition is the effect of co-exposures to metals, or other agents, on toxicokinetics. Mixtures of metals containing Pb and/or As have been observed to increase metals deposition in the liver, at least in adult animals and that co-exposures to Hg and Cd markedly reduce Hg deposition in the kidney [121]. Whether co-exposures have an influence on heavy metals in meconium is unknown, but future epidemiology studies may benefit from understanding such influences. It may be important to understand relationships among co-occurring metals before drawing conclusions about relative exposures among cohorts.
We have found several additional points of interest in this broad survey of metals in meconium. First were questions about the significance of the results reported.What is the significance of metals in meconium in terms of development and future health? What proportion of metal intake is deposited in meconium? Presumably, once metals are deposited in meconium and reach the lower intestine they are sequestered, but this is an area in need of further exploration [122]. What proportions might be deposited or circulated elsewhere (liver, kidney, brain) and result in fetal damage? More study would be needed to determine if metals found in meconium are correlated with adverse effects and to determine what concentration should raise a flag for concern.What does a high concentration of a particular metal mean? For example, does a high meconium concentration of Cu indicate high maternal intake of Cu? Or high maternal intake of a metal that competes with Cu or increases its sequestration or deposition in meconium, placenta or elsewhere?How do mixtures of metals affect transport or induce biochemical changes both from placenta to fetus and within the fetus itself?

There were methodological questions:
How much did methodological differences influence the results reported by different groups? According to sample preparation (homogenization of all meconium produced by an infant vs. procuring a small scoop) and according to analytical technique or instrument or according to statistics and reporting approach (concentration as ppb or ug/g/kg of bodyweight.

## 5. Conclusions

Data related to second and third trimester metals exposures are available from meconium samples collected from thousands of infants. At present the usefulness of the data is limited by high variability in sample collection, analytical methods and reporting styles. However, information on long-term fetal exposures are very important and these are still quite useful. It would be helpful to the scientific and medical communities if data from multiple studies could be pooled. This would help answer, confirm or refine questions originally asked by a number of study authors about metal exposures and gestational diabetes, risk of pre-term birth or birth defects, or differences in fetal metals exposures among infants from infants whose mothers lived in highly polluted or relatively pristine environments. It would also be helpful in understanding how combinations of metals influence various shared endpoints, for example, the effects of Cd, Ba and Pb exposures on hearing deficits or risk of metabolic disorders etc. That would require acceptance of a standard protocol for sample preparation and analysis for metals or groups of metals and a common use of statistical and reporting approaches.

## Figures and Tables

**Table 1 ijerph-18-01975-t001:** Description of cohorts.

Descriptor	Authors
Preterm	[18,23,25,28,29]
Full term	[18,24,25,26,30,31,32,33,34]
Identical vs. dizygotic twins	[34]
Small vs. average for gestational age	[29]
Heart defects	[7]
Male vs. Female	[35]
Polluted area	[16,30]
Gestational Diabetic	[36,37]
Non-surviving	[18]

**Table 2 ijerph-18-01975-t002:** Specification of subjects.

Study	Cu	Cr	Fe	Mn	Zn
[23]	Terms	Preemies	Preemies	Terms	Terms
[28]	Preemies	Preemies	Preemies	Preemies	Terms
[25]	Preemies, Full term, Gestational diabetes				Preemies, Full term, Gestational diabetes
[18]	Healthy, Preemies, non-surviving				Healthy, Preemies, non-surviving

**Table 3 ijerph-18-01975-t003:** Type of statistics used in each study.

Type of Statistics	Authors
Standard	[7,15,19,24,26,28,29,33,37,38,39,40]
Order	[15,18,25,27,30,32,33,36,41,42,43,44]
Both	[15,33,35,42]
Neither	[45,46,47]

**Table 4 ijerph-18-01975-t004:** Metals considered by author: most frequently analyzed.

As	[15,27,30,36,37,40,42]
Cd	[7,15,20,24,30,37,48]
Cu	[7,15,18,19,24,25,27,28,29,31,33,37,38,42]
Hg	[30,32,36,37,41,43,44,47,48]
Mg	[15,28,29,31,37,39]
Mn	[15,19,28,29,31,37,41]
Pb	[7,15,18,24,25,27,28,30,35,38,40,49,50]
Zn	[7,15,19,24,25,28,29,30,31,33,34,37,38,41,48]

**Table 5 ijerph-18-01975-t005:** Metals considered by author: less studied.

Al	[37]
Ba	[37]
Co	[15,37]
Cr	[15,28,36,37]
Fe	[15,30]
Li	[37]
Mo	[15,28,33]
Ni	[15,37]
P	[28]
Sb	[37]
Se	[51]
Sn	[37]
V	[15]

**Table 6 ijerph-18-01975-t006:** Aluminum in meconium samples by cohort (“na” = not available in reference, “X” = not calculated by author, “LOQ” = lowest observable quantity, “LOD” =lowest decile).

Author	Cohort	Mean	Stdev	Min	Median	Max	%>LOD
[37]		6.577	na	na	Na	na	na
[28]	Premature	43	61	na	Na	na	na
[28]	Term	23	23	na	Na	na	na
[30]	Industrial	X		<LOQ	3	40.7	64.5%
[30]	Not industrial	X		<LOQ	<LOQ	22.8	44.4%
[37]	Diabetic mother	32.9	103.6	na	Na	na	na
[37]	Mother not diabetic	19.4	29.9	na	Na	na	na

**Table 7 ijerph-18-01975-t007:** Cobalt in meconium.

Study	Cohort	Mean Co (ppm)	StDev
[15]	Not detailed	0.061	0.01–0.17 IQR
[51]	Term infants	1.01	0.14 StDev
[37]	Term infants	0.15	0.07 StDev
[37]	Infants of Diabetic Mothers	0.15	0.06 StDev

IQR—Interquartile Range. StDev—Standard Deviation.

## Data Availability

All data used is available in the references.

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
