# Peer review of "Evaluation of Fetal Exposures to Metals and Metalloids through Meconium Analyses: A Review"

_ijerph, 2021, doi:10.3390/ijerph18041975_

Round 1
Reviewer 1 Report
Review IJERPH ISSN 1660-4601
General comments
The manuscript review entitled “Metals and Metalloids in Meconium: A survey of the literature” and labeled IJERPH 1660-4601, is focused on presenting relevant information from the literature about metal(loid)s in meconium, highlighting their use as a prenatal exposure tool assessment. The manuscript is general professionally written and the information presented is relevant for a particular scientific community sector.
Major revisions
Why do some tables mention works just with author last names and others also include year? Please homogenize
Why do authors use element symbols in capitals, is this correct?
There are two tables 3 and two tables 4 and two tables 5, please correct numbers
Why does table 6 have no title?
Authors use sometimes “Al” others “aluminum” within the text, please homogenize, same for antimony and other elements
Minor revisions
Line 25: Please explain acronym ADHD
Line 26: “type 1 diabetes” was already mentioned in line 23
Line 50 to 51: Why is this reference underlined
Line 76: Did the authors mean “Hg” instead of hg?
Line 76: why some elements are searched by symbol and others by name? Does this make any difference?
Line 131: Did the authors mean “Gestational” instead of Gestaional?
Line 142: What is the meaning of the star in “Preemies”?
Line 181: Why is “ppm” underlined
Line187: Please explain acronym AF
Line 210: Did the authors mean “B12” instead of b12?
Line 239: Please explain acronym ONT
Line 381: Why is this reference underlined
Line 393 &403: Why is this reference underlined…
Lines 448 to 469: Some of this relevant information can be used in the abstract to highlight the content of the manuscript
Lines 471 t0 483: There are some conclusions regarding the statistical analysis?
Author Response
Major revisions
Why do some tables mention works just with author last names and others also include year? Please homogenize. We were attempting to save space by listing years in the first occurrence. We have modified to include years in all tables.
Why do authors use element symbols in capitals, is this correct? Corrected.
There are two tables 3 and two tables 4 and two tables 5, please correct numbers. Corrected
Why does table 6 have no title? Corrected
Authors use sometimes “Al” others “aluminum” within the text, please homogenize, same for antimony and other elements. Corrected
Minor revisions
Line 25: Please explain acronym ADHD. Attention Deficit Hyperactivity Disorder added
Line 26: “type 1 diabetes” was already mentioned in line 23. Text has been clarified.
Line 50 to 51: Why is this reference underlined. This was a hyperlink. Underlines have been removed.
Line 76: Did the authors mean “Hg” instead of hg? We did, but this the way we used it in the query. It has been changed to avoid confusion.
Line 76: why some elements are searched by symbol and others by name? Does this make any difference? It does because searching for “As” (Arsenic) was likely to pull in hundreds and potentially thousands of irrelevant papers. Same for “Co” and “cobalt”. We chose to use “Pb” in place of “lead” for similar reasons.
Line 131: Did the authors mean “Gestational” instead of Gestaional? Corrected
Line 142: What is the meaning of the star in “Preemies”? removed
Line 181: Why is “ppm” underlined. Corrected.
Line187: Please explain acronym AF. AF was defined earlier on line 41 as amniotic fluid.
Line 210: Did the authors mean “B12” instead of b12? Corrected.
Line 239: Please explain acronym ONT (Ontario. Abbreviation from Canadian Postal Codes). Now spelled out.
Line 381: Why is this reference underlined Hyperlink corrected
Line 393 &403: Why is this reference underlined… Hyperlink corrected
Lines 448 to 469: Some of this relevant information can be used in the abstract to highlight the content of the manuscript. Done.
Lines 471 t0 483: There are some conclusions regarding the statistical analysis? Concluding statement added on lines 160-161
Reviewer 2 Report
This paper provides an overview of the scientific literature on metals concentrations in meconium. Such exposures are a significant concern because of their potentially harmful effect on the fetus and developing child (e.g., inducing oxidative stress, altering protein kinase and glucose levels, and impacting calcium metabolism). Some metals are well known as neurotoxins. Meconium, which is the first set of stools of a newborn, represents a repository of exposures accumulated in the fetus from the 12th week of gestation until birth. This technique of collection is relatively noninvasive, inexpensive and tends to have sensitivity and specificity regarding environmental exposure. Overall, this is a well-written and organized manuscript and provides an excellent overview on the topic. The reference search was comprehensive and up to date. If relevant, my only suggestion is to add a limitations paragraph to the manuscript.
Author Response
Thank you for your kind review. We have included a discussion of limitations in the discussions of statistics and methods.